# Mechanics of lung cancer: A finite element model shows strain amplification during early tumorigenesis

**Rebecca G. Zitnay**[1,2], **Michael R. Herron**[1], **Keith R. Carney**[3], **Scott Potter**[2,4], **Lyska L. Emerson**[2,4], **Jeffrey A. Weiss**[1,5]*, **Michelle C. Mendoza**[1,2,3]*

**1** Department of Biomedical Engineering, University of Utah, Salt Lake City, Utah, United States of America, **2** Huntsman Cancer Institute, Salt Lake City, Utah, United States of America, **3** Department of Oncological Sciences, University of Utah, Salt Lake City, Utah, United States of America, **4** Department of Pathology, University of Utah, Salt Lake City, Utah, United States of America, **5** Scientific Computing and Imaging Institute, Salt Lake City, Utah, United States of America

* Jeff.Weiss@utah.edu (JAW); Michelle.Mendoza@hci.utah.edu (MCM)

**Data Availability Statement:** The authors confirm that all data underlying the findings are fully available without restriction. The MatLab code used to construct the lung lattice geometry has been uploaded to a publicly available repository: (https://

## Abstract

Early lung cancer lesions develop within a unique microenvironment that undergoes constant cyclic stretch from respiration. While tumor stiffening is an established driver of tumor progression, the contribution of stress and strain to lung cancer is unknown. We developed tissue scale finite element models of lung tissue to test how early lesions alter respiration-induced strain. We found that an early tumor, represented as alveolar filling, amplified the strain experienced in the adjacent alveolar walls. Tumor stiffening further increased the amplitude of the strain in the adjacent alveolar walls and extended the strain amplification deeper into the normal lung. In contrast, the strain experienced in the tumor proper was less than the applied strain, although regions of amplification appeared at the tumor edge. Measurements of the alveolar wall thickness in clinical and mouse model samples of lung adenocarcinoma (LUAD) showed wall thickening adjacent to the tumors, consistent with cellular response to strain. Modeling alveolar wall thickening by encircling the tumor with thickened walls moved the strain amplification radially outward, to the next adjacent alveolus. Simulating iterative thickening in response to amplified strain produced tracks of thickened walls. We observed such tracks in early-stage clinical samples. The tracks were populated with invading tumor cells, suggesting that strain amplification in very early lung lesions could guide pro-invasive remodeling of the tumor microenvironment. The simulation results and tumor measurements suggest that cells at the edge of a lung tumor and in surrounding alveolar walls experience increased strain during respiration that could promote tumor progression.

## Author summary

Lung cancer is the leading cause of cancer-related death in the world. Efforts to identify and treat patients early are hampered by an incomplete understanding of the factors that

github.com/MendozaLabHCI/
TumorLungMechanics_2022). FEBio and
FEBioStudio are open-source software projects.
Installation packages and source code are freely
available at https://febio.org/. The bulk lung
constitutive model from Birzle et. al. was
implemented in the release versions of FEBio and
FEBioStudio.

**Funding:** The research reported in this publication
was supported by the National Institutes of Health
under Award Number P30CA042014, and
R01GM083925 (JAW), U24EB029007 (JAW), and
R01CA255790 (MCM), as well as by the American
Cancer Society RSG CSM130435 (MCM). The
content is solely the responsibility of the authors
and does not necessarily represent the official
views of the NIH.The funders had no role in study
design, data collection and analysis, decision to
publish, or preparation of the manuscript.

**Competing interests:** The authors have declared
that no competing interests exist.

drive early lesion progression to invasive cancer. We aimed to understand the role of mechanical strain in early lesion progression. The lung is unique in that it undergoes cyclic stretch, which creates strain across the alveolar walls. Computational models have provided fundamental insights into the stretch-strain relationship in the lung. In order to map the strain experienced in the alveolar walls near a tumor, we incorporated a tumor into a tissue scale model of the lung under stretch. We used finite element modeling to apply physiological material behavior to the lung and tumor tissue. Based on reported findings and our measurements, tumor progression was modeled as stiffening of the tumor and thickening of the tumor-adjacent alveolar walls. We found that early tumors amplified the strain in the tumor-adjacent alveolar walls. Strain amplification also arose at the tumor edges. Simulating strain-mediated wall stiffening generated tracks of thickened walls. We experimentally confirmed the presence of tracks of thickened extracellular matrix in clinical samples of LUAD. Our model is the first to interrogate the alterations in strain in and around a tumor during simulated respiration and suggests that lung mechanics and strain amplification play a role in early lung tumorigenesis.

## Introduction

Lung cancer is the primary cause of cancer-related death worldwide [1]. New screening efforts aim to catch pre-cancerous and early-stage disease, when a surgical cure is possible [2]. However, even after surgical resection and treatment, 23% of patients with stage I or stage II disease suffer a local recurrence within 5 years [3]. Improving clinical management requires a better understanding of the factors that contribute to early lung tumor progression, including changes within the tumor microenvironment.

Lung adenocarcinoma (LUAD) is the most common form of lung cancer. LUAD is initiated by oncogenic mutations in lung epithelial cells that line the alveoli. The mutations cause aberrant cell proliferation, causing transformed cells to fill the alveolar airspace and thicken the alveolar walls [4]. Oncogenic *KRAS* is the most common initiator of LUAD and accounts for ~30% of disease [5,6]. Additional mutations, such as loss of tumor suppressor TP53, and alterations in the tumor microenvironment contribute to the tumor's progression to metastatic cancer [5,7,8]. The deposition of extracellular matrix (ECM) proteins like Tenascin-C and crosslinking of collagen promote metastasis and correlate with poor prognosis [9,10]. These ECM changes often accumulate into a desmoplastic stroma [11,12], which is associated with tumor stiffening and aggressiveness in many solid tumors [13–17].

Lung tumors are unique in that they develop in a soft tissue environment that undergoes regular cyclic stretch from respiration. The contribution of mechanical strain to early lung tumorigenesis is unknown. Alveolar surface area increases ~50% at total lung capacity, reached during deep inspiration or ventilation [18,19]. Live imaging has shown that alveolar segments elongate ~15% at inflation volumes near total lung capacity [20]. Both lung epithelial and stromal cells sense and respond to strain [21–23]. Strain can induce oncogenic cellular processes, such as growth factor and matrix production and cell proliferation and migration [24–30]. During lung injury, strain signaling for ECM deposition and cell proliferation drives tissue repair [31]. The process becomes pathological in pulmonary fibrosis. In this disease, persistently elevated strain creates a signaling loop for continuous cell proliferation and ECM deposition [32]. Fibrosis-associated scar tissue and alveolar wall thickening impair lung elasticity and lung function [33,34]. The desmoplastic stroma of late-stage LUAD resembles fibrosis [9,35], suggesting that elevated strain may play a role in LUAD pathogenesis.

We hypothesized that early tumors locally amplify the strain in surrounding alveolar walls during respiration. Since direct measurement of localized strains in the intact lung is experimentally challenging, we mapped the strain environment around a lung tumor by computationally modeling the mechanics of the alveolar network. In order to determine the stress-strain behavior of individual alveoli in lung tissue containing a tumor, we developed a finite element model of the alveolar network. Previous lung models have characterized the material behavior and geometric distortions of the lung during respiration and fibrosis. Early constitutive lung models described the uni-axial stretch of bulk lung tissue [36–38]. Early network models with lattices of spring elements revealed the relationship between lung pressure and volume [39,40]. They showed that alveolar wall stress and stiffness control lung elasticity [39,40]. More recently, a geometric study of lung distortion showed that uni-axial stretch causes the alveolar walls to align parallel to the axis of stretch, rather than elongate [41]. The more physiological elongation-type deformations were captured by new constitutive models that applied bi-axial stretch to decellularized tissue [42] and incorporated pressure-volume changes in bulk lung [43]. While the model by Birzle et al. described native lung behavior [43], it did not allow for the investigation of heterogeneous alveoli deformation.

We developed a tissue-scale finite element model of the lung using FEBio [44]. Previous continuum constitutive models characterized the heterogeneity in cell and alveoli responses to strain, but were based on uni-axial stretch experiments [45,46] or decellularized tissue [47]. Our model applies the physiological stress-strain relationship of lung tissue [43] to a network of a randomized hexagonal lattice of 3D shell elements [48]. The alveolar walls can undergo elongation, bending and shear. A tumor was incorporated in the center of the lattice using 3D solid elements. When we applied biaxial stretch to simulate physiological tension, we found that the tumor amplified strain in the alveolar walls adjacent to the tumor and in the tumor edge. Modifying the tumor to simulate the increased stiffness associated with late-stage disease resulted in greater strain amplification in alveolar walls. Measurements of alveolar walls in LUAD samples from mouse models and clinical cases showed that tumor-adjacent walls were thickened. When we incorporated thicker alveolar walls into our lung model, the thickening resulted in strain amplification that persisted farther away from the tumor radially. Iterative strain-dependent thickening generated tracks of thick walls. We identified similar tracks of ECM deposition at the invasive edge of human LUAD, suggesting that early lung lesions create local strain amplification that could contribute to tumor progression.

## Methods

### Ethics statement

Human data collected and analyzed for this research project was approved by the University of Utah Institutional Review Board (approval #00141909I/F and 89989). Prior to obtaining tissue for analysis, all samples were de-identified to comply with HIPAA regulations. The staining and analysis of slides were carried out in accordance with relevant guidelines and regulations.

### Characterization of lung cancer geometry and ECM in a mouse model of lung cancer

Early LUAD presents as filled alveoli or lepidic growth, in which tumor cells expand along and thicken the alveolar walls [4]. Mouse models have the advantage of capturing the earliest phase of tumor formation, before clinical detection would be possible [6,8]. We confirmed the geometry of early LUAD lesions in tumors from a mouse model of KRAS-driven LUAD with concomitant silencing of TP53. Under University of Utah IACUC # 18–08005 and #21–10007,

**Fig 1. Early lung adenocarcinoma (LUAD) is characterized by alveolar filling.** Lung tissue slides stained with Masson's Trichrome to distinguish cells from connective tissue. Cell nuclei are dark purple, cytoplasm light red, and collagen blue. **A.** Alveolar architecture of normal lung from $Kras^{+/+}$; $Trp5^{f/f}$ mice. **B.** Tumor from $Kras^{LSL-G12D/+}$; $Trp53^{f/f}$ mice 10 weeks and **C.** 25 weeks after intra-tracheal inoculation with SPC-Cre. Images are representative of lung tissue from 3 $Kras^{+/+}$ mice or tumors from 3 mice $Kras^{LSL-G12D/+}$ mice at each time point.

$KRas^{+/+};Trp53^{f/f}$; $TdTomato$ control mice and $KRas^{LSL-G12D/+};Trp53^{f/f}$; $TdTomato$ mice were intratracheally infected with $1x10^8$ pfu/mouse SPC-Cre Adenovirus (University of Iowa Viral Vector Core). Cre-mediated recombination in lung epithelial cells induces $Trp53$ deletion and tdTomato expression in control mice and simultaneous $KRAS^{G12D}$ mutation, $Trp53$ deletion, and tdTomato expression in experimental mice. After 10 weeks of tumor growth (Early), the animals were anesthetized with $CO^2$ and the lungs were perfused with 10% neutral buffered formalin to a normal inflation volume. Formalin-fixed lungs were paraffin-embedded and sectioned to 5 μm by the Biorepository and Molecular Pathology shared resource. Slides were stained with Masson's Trichrome (Sigma) to image the cells and ECM, following the manufacturer's protocol. Slides were imaged on a Pannoramic Midi II slide scanner and analyzed with associated CaseViewer software (3D Histech). As previously reported, $Kras^{+/+}$ mice did not have any tumors and the alveoli formed a lattice of hexagonal-like units (Fig 1A, [8]). Transformed cells harboring mutant KRAS$^{G12D}$ developed into small early tumors that appeared as filled alveoli (Fig 1B). We also confirmed the presence of desmoplastic stroma in advanced tumors grown for 25 weeks (Late). Compared to the early tumors, the late tumors were larger and exhibited a more compact cell organization and increased collagen deposition (Fig 1B and 1C, [8,9]). Images are representative of $n$ = 3 mice in each group.

## Description of the finite element models of lung tissue

We created two computational models of lung tissue, one lacking and one harboring a tumor. For both the normal and tumor models, we created an idealized geometric finite element model by representing the alveolar architecture with a mesh of a hexagonal lattice of 6-noded 100 x 100 units in MATLAB (Mathworks, Matick, MA) (Fig 2A). Each alveolus measured ~100 μm in diameter, which is similar to the dimension of alveoli in the human lung at physiological pressure [21]. To account for variability in lung architecture, a randomization algorithm was applied to the initial geometry, followed by simulated annealing through energy minimization [48]. Briefly, the 6 nodes of the hexagonal units were subjected to a random displacement (alpha = 0.1) followed by rotation of the resulting hexagons (beta = 4.21) such that the resulting aspect ratio was ~0.2. MATLAB software for the randomization is shared via GitHub at https://github.com/MendozaLabHCI/TumorLungMechanics_2022. The randomized lattices were imported to FEBio Studio [44], open-source and available at https://febio.org/. The FEBio Studio platform readily solves non-linear large deformation problems and allows material properties to be assigned to

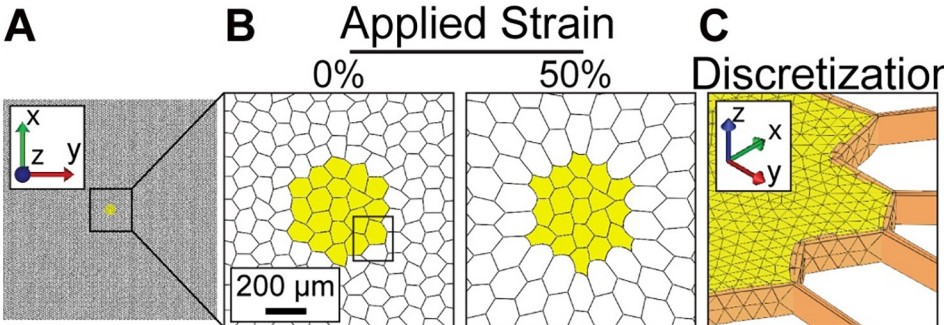

**Fig 2. Implementation of a geometric tissue-scale constitutive model of lung cancer. A.** Lung model, in which alveoli are represented as shell elements organized in a randomized hexagonal lattice. Model is 9,885 μm x 10,000 μm x 30 μm. **B.** Zoomed-in view of tumor and adjacent matrix under 0% and 50% globally applied stretch in x and y, displacement in z is fixed. **C.** Discretization of the lung model. Alveoli are represented by shell elements. The tumor is discretized using tetrahedral elements with a surrounding basement membrane of triangular elements.

individual components of a complex geometry. A finalized lattice had 100 hexagonal units (9,885 μm) in the x-direction and 100 units (10,000 μm) in the y-direction and was modeled with a z-height of 30 μm. Shell elements with thickness of 10 μm represented the alveolar walls. This discretization method more faithfully represented the geometry and material properties of alveoli than 1D discrete elements used in previous models of alveoli [39,40,48].

For the tumor model, a tumor was incorporated as 20 filled alveolar units in the center of the lattice (Fig 2A and 2B), consistent with early tumors found in *KRas*<sup>*LSL-G12D/+*</sup>*;Trp53*<sup>*f/f*</sup>*; TdTo-mato* mice. The tumor was discretized as 11,829 solid 4-noded tetrahedral elements (Fig 2C). A mesh convergence study was performed. We tested the order (1<sup>st</sup> order linear vs. 2<sup>nd</sup> order quadratic behavior) and the number of elements to refine the mesh under 50% bi-axial elongation stretch. We found that the number of elements, but not the order, led to an increased tumor diameter, up to 11,829 elements (S1 Table). A basement membrane surrounding the tumor was discretized as triangular elements that connect the tumor to the adjacent lung matrix.

The finite element models were subjected to equibiaxial elongation in increments of 5% and up to 50% elongation. Stretch was applied on the 4 boundaries in x and y to obtain the prescribed elongation. Plane strain boundary conditions were assigned to both the non-tumor and tumor-containing models. Thus, equibiaxial elongation was applied along the x and y directions while their geometry was constrained in the z-direction. With stretch, deformations of the alveolar walls and tumor were evident (Fig 2B).

### Parameter optimization based on bulk lung constitutive model

To ensure that the constitutive model and associated material coefficients that we used for the alveolar walls provided an accurate representation of lung tissue material behavior, we implemented a hyperelastic constitutive model in FEBio that was developed previously by Birzle et al. to describe the continuum level behavior of bulk lung tissue [43]. To determine the material behavior of the alveolar walls in our lattice geometry, we fit the material coefficients of a neo-Hookean hyperelastic constitutive model to the behavior predicted by the Birzle constitutive model.

The strain energy density for the Birzle hyperelastic constitutive model is defined as:

$$W = c(I_1 - 3) + \frac{c}{\beta}\left(I_3^\beta - 1\right) + c_1\left(I_3^{1/3}I_1 - 3\right)^{d_1} + c_3\left(I_3^{1/3} - 1\right)^{d_3} \tag{1}$$

Here, $I_1$ and $I_3$ are the invariants of the right Cauchy-Green deformation tensor $\boldsymbol{C}$, and $c$, $\beta$, $c_1$, $c_3$, $d_1$ and $d_3$ are material coefficients. The elastic modulus and Poisson's ratio of this material

are related to the material coefficients:

$$E = 4c(1 + v), v = \frac{\beta}{1 + 2\beta} \tag{2}$$

This material has been shown to faithfully represent the tissue level stress-strain response of alveolar tissue.

The compressible neo-Hookean hyperelastic constitutive model in FEBio that we used to represent the alveolar walls has strain energy:

$$W = \frac{\mu}{2}(I_1 - 3) - \mu \ln(J) + \frac{\lambda}{2}(\ln J)^2, \tag{3}$$

where $J$ is the determinant of the deformation gradient $F$, and $\lambda$ and $\mu$ are the Lamé coefficients, which can be related to $E$ and $v$:

$$\mu = \frac{E}{2(1 + v)}, \lambda = \frac{vE}{(1 + v)(1 - 2v)} \tag{4}$$

We prescribed $v = 0.25$ to represent moderate tissue compressibility and determined $E$ by material parameter optimization to match the behavior of bulk lung tissue under uniaxial tension and air expansion, ~2 kPa [43]. To do this, we developed a geometric lung lattice of 100 hexagonal units (10 mm) in the x-direction and 30 units (3.3 mm) in the y-direction, which matches the size of a rectangular prism assigned to the constitutive model from Birzle, et al. [43]. We then optimized the force-displacement behavior of the lattice using a prescribed global uniaxial elongation of 50%. The relationship between force and displacement of our test lattice and the Birzle rectangular prism [43] was approximately linear for displacement values of 0–15,000 μm (0–50% strain), consistent with previous findings that demonstrate an approximately linear stress-strain relationship for bulk lung within the physiological strain range (S1 Fig, [48–50]). The neo-Hookean constitutive model was assigned to the lung lattice model. The resulting optimized modulus for the alveolar walls of the normal lung was $E = 35$ kPa. We noted that this modulus is between unstretched and stretched tissue [42], consistent with a system under prestress and as expected, is greater than measurements in tissue slices 40 times thicker [34].

To corroborate our approach, we tested that our neo-Hookean constitutive model could be applied to predict an effective modulus that was consistent with atomic force microscopy (AFM) and microindentation measurements of human and pig lung tissue, ~1.8 kPa (Table 1, [38,51]). We simulated AFM indentation in FEBio using a 30 μm thick finite element mesh (S2 Fig). The resulting prediction for reaction force on the indenter, $F = 8.2 \times 10^{-9}$ N, was used to solve for the material modulus ($E$) by the Hertz model (Eq 5):

$$E = \frac{3F(1 - v^2)}{4(\sqrt{Rd^3})} \tag{5}$$

where $v = 0.25$, the indenter radius $R = 2.5$ μm, and the indentation depth $d = 2.5$ μm. This yielded $E = 0.9$ kPa, which is on the order of experimental measurements [38, 51].

Lung tumors were represented with the neo-Hookean constitutive model using $v = 0.49$ [15]. We represented early LUAD with $E = 2$ kPA, matching the modulus of the bulk lung tissue (Fig 3B) and we represented advanced LUAD with $E = 20$ kPA (Fig 3C). We used tumor modulus measurements from brain and breast tissue, the two closest organs in modulus, as benchmarks [52]. In these tissues, tumors have modulus that is ~10 times greater than normal tissue (Table 2, [15,16]). This relative difference is also consistent with the difference in measured modulus values between fibrotic lung tissue and normal lung tissue [34].

**Table 1. Reported Mechanical Properties of Lung Tissue[a].**

| Tissue Source | Measurement Mode | Preparation | E Healthy (kPa) | E Fibrotic (kPa) | $v$ | Reference |
|---|---|---|---|---|---|---|
| Human | AFM, spherical tip, radius = 2.5 μm, k ~0.06 N/m | frozen and thawed, 10 μm thick | 1.87 ± 0.95 | N/A | 0.4 | [51] |
| Pig | microindentation<br>uniaxial Tension | fresh, 1.6–2.7 mm thick<br>fresh, 2.7 cm x 1 cm x 2 mm thick | 1.4 ± 0.4<br>3.4 ± 0.4 | N/A | 0.42 | [38] |
| Rat | uniaxial tension and volume pressure change | gelatin inflated, 5–9 mm x 4–8 mm x 0.5–2 mm thick | 1.91 ± 0.13 | N/A | 0.34 | [43] |
| Rat | uniaxial tension<br>AFM indentation, under equibiaxial tension | decellularized and stored at 4˚C; ~7 mm x 2 mm x 2 mm thick<br>decellularized, frozen, and thawed; 16.8 μm thick | unstretched: 0.38 ± 0.01; stretched (20%): ~2.5<br>unstretched: 7.9 ± 0.7; stretched (20%): ~50 | N/A | 0.5 | [42] |
| Mouse | AFM, spherical tip, radius = 2.5 μm, k ~0.06 N/m | agarose inflated, 5mm x 5mm x 400 μm thick. | ~0.5 | ~3 ± 2 | 0.4 | [34] |

[a] Measurements of lung parenchyma. AFM experiments are limited to those with tips functionalized for the measurement of tissue modulus, rather than fiber modulus.

(E) is modulus, as mean ± standard deviation when reported.

($v$) is Poisson Ratio.

## Calculation and representation of effective strain

We mimicked the global strain that results from respiration by subjecting the final discretized mesh to equibiaxial elongation, applied at the boundaries. Plane strain boundary conditions fixed the deformation in the z-direction. We applied equibiaxial strains of 5%, 20%, and 50%, which represented the range of normal respiration (5–20%, [20]), and maximum elongation (50%, [53]). We assessed the relationship between applied global strain and local strain by calculating the ratio. The applied strain induced shear strain and first principal strain. We focused on the local first principal strain to understand how the alveolar walls stretch in the absence and presence of a tumor. To find a ratio of comparable quantities, we converted the applied infinitesimal strain $\varepsilon$, to the Green-Lagrange strain $E_A$:

$$E_A = \frac{1}{2}(\varepsilon^2 - 1). \tag{6}$$

We then calculated the ratio of local Green-Lagrange strain $E_L$, with the applied strain $E_A$.

$$strain\ ratio = \frac{E_L}{E_A} \tag{7}$$

We created fringe plots that display the strain within surface elements of the tumors. Internal elements can have strain amplification greater or less than displayed values and are included within strain ratio calculations. When comparing the strain ratio in alveolar elements adjacent to a tumor to alveoli in normal tissue, the tumor boundary was mapped onto the normal tissue so that the corresponding tumor-adjacent alveolar elements in the normal lung were assessed.

## Statistics

For comparisons between groups, the data sets were tested for normality using the Anderson-Darling test (α = 0.5). Not all data sets were normally distributed, so non-parametric tests were employed. For comparison of mean alveolar wall widths, significance between the two groups was tested using the Mann-Whitney test. For comparison of mean strain ratios, significance

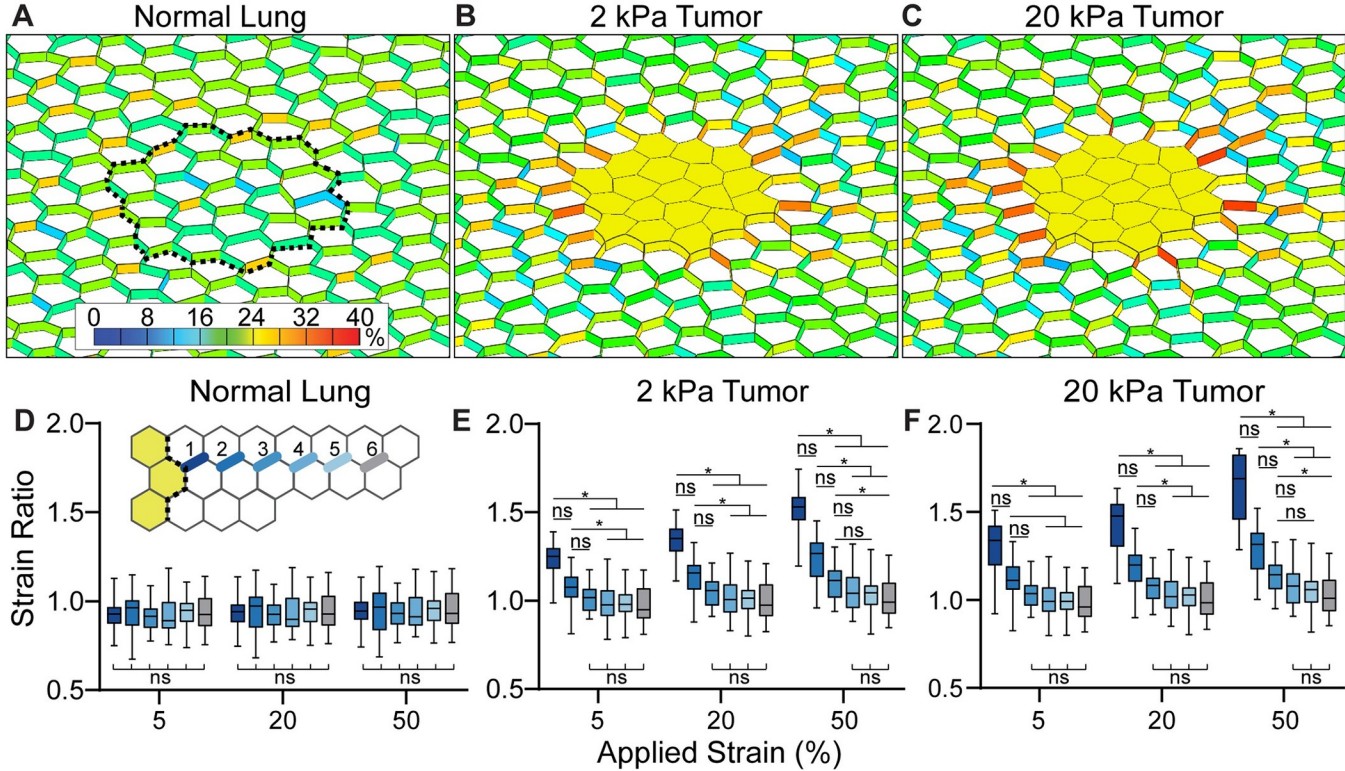

**Fig 3. Lung tumors amplify strain in surrounding alveoli during inspiration.** Strain colormaps after 20% stretch. **A.** Normal lung, **B.** lung with 2 kPa early tumor, and **C.** lung with a 20 kPa late tumor. Dotted line represents the location on the normal mesh of the tumor boundary. Colormap is % Green-Lagrange shell strain. **D-F.** Strain ratios in each model. Inset in D. is a schematic of the measured alveolar wall segments in which the dark-to-light colors represent the distance from the tumor edge. Boxes are 25th percentile to 75th percentile with the median and maximum and minimum values labeled. In D., for all comparisons of strain ratio in the normal lung, $p > 0.999$. Distribution includes all alveolar wall segments at each indicated distance.

was tested with the Kruskal-Wallis test, which allows comparisons of three or more means. To determine which means were different from each other, a post-hoc Dunn's multiple comparisons test was used. (*) indicates $p < 0.05$, which was considered significant.

**Table 2. Reported Mechanical Properties of Tumor Tissue.**

| Tissue source | Mode of Measurement | Tissue Preparation | E Normal/Healthy (kPa) | E Diseased/Tumor (kPa) | ν | Reference |
|---|---|---|---|---|---|---|
| Mammary (Mouse) | electrochemical indentation | fresh | 0.167 ± 31 | Tumor: 4.049 ± 0.938 Adjacent Stroma: 0.918 ± 0.269 | | [13] |
| Breast (Human) | AFM, spherical tip, radius = 2.5 μm sphere, k ~0.06 N/m | frozen and thawed, 20 μm thick | 0.4 | Tumor: 2–8 | 0.5 | [15] |
| Breast (Human) | AFM, pyramidal tip, radius = 20 nm, k ~0.06 N/m | fresh, biopsy diameter = 0.2 cm, length 0.1 = 1 cm | Breast: 1.130 ± 0.780 Lung: 1.000 ± 0.500 | Primary (bimodal distribution): 0.450 ± 0.150 & 1.260 ± 0.430 Metastasis to the lung: 0.56 ± 0.26 | | [79] |
| Brain (Human) | AFM, with spherical tip, radius = 2.5 μm sphere, k ~0.06 N/m | frozen and thawed, ~30 μm thick | 0.010–0.018 | LGG (grade I& II): 0.050–1.400 GBM (grade IV): 0.070–13.500 | | [16] |

LUAD: Lung adenocarcinoma

LGG: Low Grade Glioma

GBM: Glioblastoma

### Alveolar wall width measurements and modeling alveolar wall thickening

The width of the alveolar wall was measured within 3 alveoli from the tumor edge and at the narrowest region along the length of an alveolar wall segment between junctions. We measured the entire distinguishable edge: 18–37 alveolar walls per tumor.

We calculated alveolar wall thickness in 3 mouse tumors from $KRas^{LSL-G12D/+;}Trp53^{f/f}$; TdTomato mice 10 weeks after induction (described in *Characterization of lung cancer geometry*), stained by Masson's Trichrome. Immunohistochemistry (IHC) for TdTomato (1:1200, Rockland, Cat. 600-401-379) in serial sections labeled the tumor. As a control, alveolar wall thickness was measured in uninvolved regions lacking TdTomato and at least 1 mm away from the tumor edge.

We examined 5 cases of human T1 LUAD and 2 areas of human uninvolved lung, defined as > 4cm away from any suspected tumor. Under IRB# 00141909I/F and #89989 at the University of Utah, deidentified human samples were obtained from lung cancer resections. Small (< 3 cm T1 and < 4 cm T2) LUAD cases were selected for study. Formalin-fixed, paraffin-embedded human tumor sections were stained with hematoxylin and eosin (H&E) by the HCI Biorepository and Molecular Pathology shared resource. Hematoxylin labels tumor cell nuclei dark blue-purple. Eosin labels other structures light pink. The tumor border was marked by a board-certified pathologist.

To mimic the alveolar wall thickening observed in tumor-adjacent tissue, wall segments in the first layer of adjacent alveoli were set to a thickness of 30 μm.

### Progressive alveolar thickening

To test the consequence of progressive alveolar wall thickening, a strain-mediated alveolar wall thickening simulation was run iteratively. For each iteration, the width of any shell element with a strain magnitude that exceeded a threshold value of 28% was increased to 30 μm in a circular region of interest that included 1,064 alveolar wall elements around the tumor area. For each iteration, shell thickness and strain were mapped (S4 Fig).

To visualize matrix deposition around the tumor, human tumor slides were stained by immunohistochemistry with an antibody against Tenascin-C (1:300, EMD Millipore, Cat. ab19011), a strain-sensitive ECM protein known to be present in LUAD [9, 54]. Briefly, sections were deparaffinized and rehydrated. IHC was performed manually using the Vectastain ABC Kit (Vector Laboratories). Target retrieval was performed in citrate buffer at pH 6.0 for 15 minutes at 110°C under pressure and sections were transferred to Sequenza slide staining racks (Electron Microscopy Sciences). Sections were treated with Bloxall (Vector labs) followed by horse serum (Vector Labs, Burlingame, CA), anti-Tenascin-C primary antibody overnight at 4°C, and HRP-polymer-conjugated secondary antibody (anti-rabbit from Vector Labs). The slides were developed with Impact DAB (3,3'-Diaminobenzidine, Vector) and counterstained with hematoxylin.

## Results

### Strain in the alveoli around the tumor

To understand the distribution and intensity of strain in the early lung tumor environment, we mapped the strain expected during expansion in our models of normal and tumor-containing lung tissue. In the normal lung model, 20% applied strain resulted in a median first principal Green-Lagrange strain of 20.51% in the alveolar walls (Figs 3A and S3A). As expected, the randomized geometry resulted in small variations in the alveolar wall strain [48] that were uniformly distributed across the lung (Figs 3A and S3A). In the early tumor (2 kPa), applying a

20% strain resulted in a median Lagrange strain of 29.8% in the alveoli directly adjacent to the tumor (Figs 3B and S3B). The strain decreased to ~20% in the alveoli further from the tumor (Fig 3B). The amplification was most notable in the alveolar walls perpendicular to the tumor edge, was greater in some walls than others, and reached a maximum strain of 33.3% (Fig 3B). In the late tumor (20 kPa), the median strain in the tumor-adjacent alveolar walls was 32.5% (Figs 3C and S3B). As in the early tumor, the localized strain amplification in the 20 kPa tumor decreased with distance from the tumor to ~20%.

We calculated the strain ratio (Eq 7) as a function of applied stretch and distance from the tumor. The application of 5%, 20%, and 50% stretch reflected the regional tissue strain expected during normal respiration (5–20%) and at maximum tidal volume (~50%) (Fig 3D–3F). There was no difference between the strain ratio in normal lung when applied stretch or distance from the tumor was increased (Figs 3D and S3B). In contrast, in both early and late tumor conditions, the amount of strain amplification increased with applied stretch and decreased with distance from the tumor (Figs 3E, 3F and S3B). The median strain ratio in the alveoli bordering a 2 kPa tumor was increased from 1.25 with 5% applied stretch to 1.53 with 50% applied stretch (S3B Fig). In the 20 kPa tumor, the median strain ratio in the alveoli bordering the tumor was increased from 1.34 with 5% applied stretch to 1.69 with 50% applied stretch (S3B Fig). In all cases, the strain ratio decreased to ~1 within 3 alveolar units away from the tumor (Fig 3E and 3F).

Since tumor modulus changes during tumor progression, we tested how modulus influenced strain in the tumor-adjacent alveoli. We varied the tumor modulus from 0.2 kPa to 35 kPa. At low modulus values ($E < 3.5$), the strain ratio increased with tumor modulus at all applied strain conditions (S3C Fig). However, at each applied strain, when $E > 3.5$, further increasing $E$ did not affect the strain ratio ($<5\%$ difference from strain ratio at $E = 35$ kPa) (S3C Fig).

## Strain within the tumor

We next examined the localized strain within the tumor proper. When subjected to 20% strain, the overall strain within the tumor was less than that of the surrounding alveoli shown in Fig 3, with a median Lagrange strain of 4.9% for the 2 kPa tumor and 1.1% for the 20 kPa tumor (Fig 4A and 4B). Fringe plots of the strain showed that the regions of strain amplification were most pronounced at the tumor edges and that amplification only happened in the early, 2 kPa tumor (Fig 4A and 4B). To find the depth of amplification from the early tumor edge, we plotted the strain ratio in the 2 kPa tumor as a function of the distance from the center of mass at 20% applied stretch (Fig 4C). We found that strain amplification occurred only in elements in the outermost 100 μm of the tumor, equivalent to ~5 LUAD epithelial cell diameters. We designated elements with a strain ratio $> 1$ "amplified elements." Amplified elements reach a maximum strain of 45.0%, which corresponds to strain ratio 2.045 (Fig 4C). We found that the amplitude of strain amplification increased with strain (Fig 4D). Since no strain amplification was observed in the late 20 kPa tumor, we sought to determine how tumor modulus influenced the presence of amplified elements. Across the range of tumor stiffness values, $E = 0.2$–20 kPa, we found that the percent of tumor elements with strain amplification was inversely correlated with stiffness (Fig 4E). The 0.2 kPa tumor exhibited 20% of elements with amplified strain, the 2 kPa tumor exhibited 0.5% of elements with amplified strain; tumors with $E < 5$ kPa did not exhibit amplified elements. As we estimate the physiological range of tumor stiffness to be $> 2$ kPa, these results suggest that the impact of strain on tumor progression is likely to occur in alveolar walls surrounding the tumor.

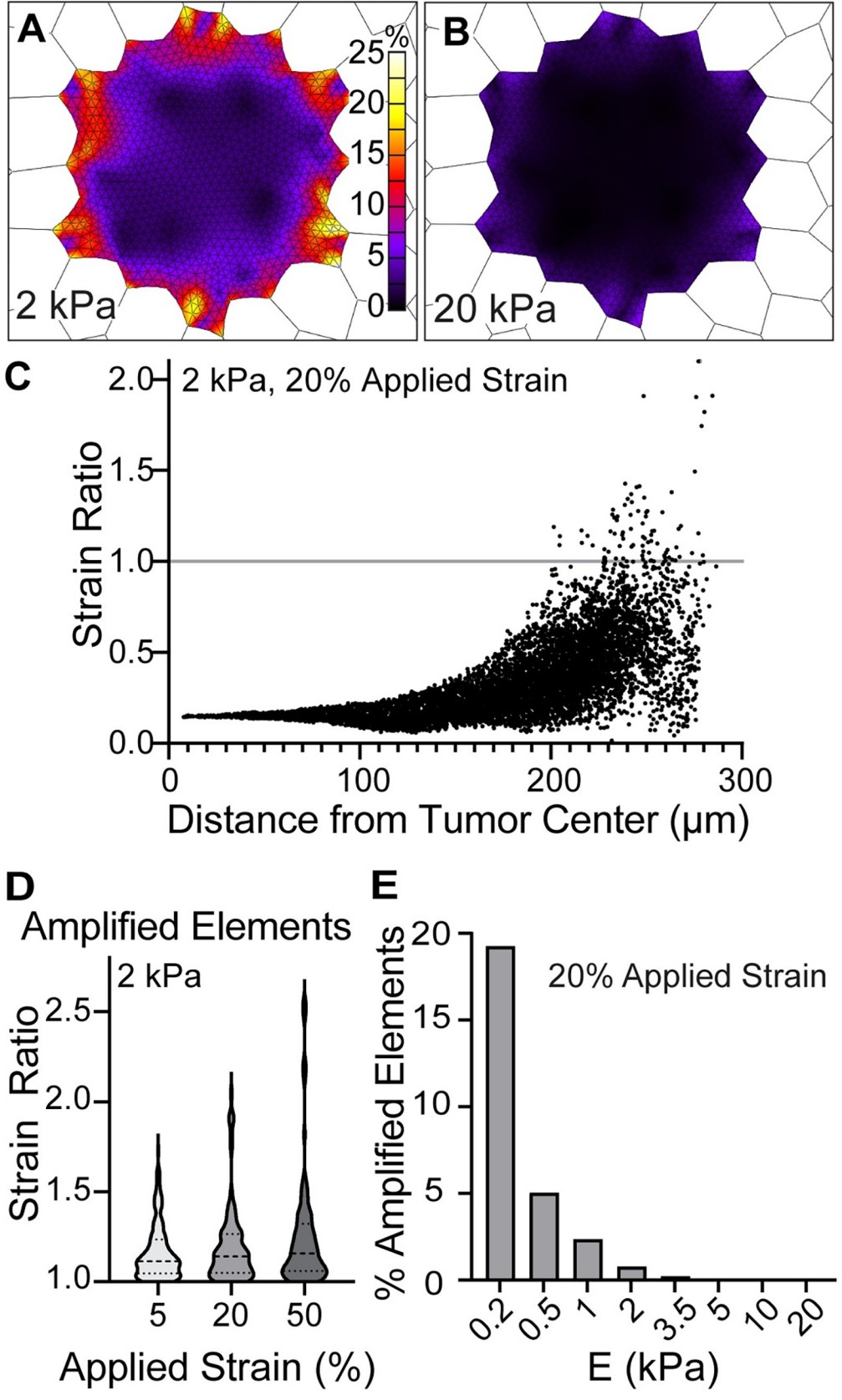

**Fig 4. Early lung tumors harbor strain amplification at the tumor edge.** Colormaps of strain after 20% stretch within the tumor in tissue with **A.** 2 kPa tumor and **B.** 20 kPa tumor. Colormaps display strain only on the tumor surface. **C.** Element strain ratio as a function of distance from the tumor center of mass, 2 kPa tumor at 20% applied stretch. Elements with strain ratio > 1 (grey line) were considered amplified. **D.** Distribution of the strain ratio in amplified elements ($n$) as a function of applied stretch, 2 kPa tumor. $n = 155$ at 5% stretch, 91 at 20% stretch, and 99 at 50% stretch, out of 11,829 elements. **E.** Percent of elements with a strain ratio >1, dependent on the tumor modulus value ($E$).

## Alveolar wall thickening around the tumor extends the area of strain amplification

We sought to incorporate the effects of tumor microenvironment ECM remodeling into our tumor models to understand how remodeling affects strain. Given that ECM deposition during fibrosis results in alveolar wall thickening [33], we tested if the tumor-adjacent alveolar walls are thickened in early lung tumors. We measured the thickness of alveolar walls adjacent to the tumor in early mouse LUAD and human tumors. In the $KRas^{LSL-G12D/+}$;$Trp53^{f/f}$; tdTomato early tumor model of LUAD, we found the wall thickness to double in width from 2.7 μm in alveoli distant from the tumor to 4.9 μm in the first 3 alveoli from the tumor boundary (Fig 5A and 5B). We confirmed the presence of matrix thickening in early, T1 human LUAD. Alveolar wall thickness increased from 8.2 ± 2.6 μm in alveoli distant from the tumor to 31.9 ± 12.9 μm adjacent to the tumor (Fig 5C and 5D).

To evaluate how a change in alveolar wall thickness would influence the strain in the tumor microenvironment, we increased the alveolar wall thickness to 30 μm in the alveoli directly adjacent to the tumor (Fig 5E). This resulted in a decrease in the mean strain ratio in the thickened alveolar walls, from 1.4 to 0.4 (Fig 5F and 5G). However, the strain ratio in the alveoli just outside the thickened region increased to a median ratio of 1.3, equivalent to that in the walls adjacent to the unmodified tumor with 10 μm wide walls (Fig 5G). These data indicate that when the matrix is thickened around the tumor, it undergoes less deformation in response to applied strain and increases the distance into the tumor microenvironment that the strain amplification reaches.

## Alveolar wall thickening around the tumor extends the area of strain amplification

Strain-mediated reinforcement of alveolar walls through increased matrix deposition is a characteristic of progressive pulmonary fibrosis [32]. Since we observed alveolar wall thickening in lung cancer samples and found that such thickening extends the amplified strain into the tumor microenvironment (Fig 5), we hypothesized that strain-mediated reinforcement may also occur in lung cancer. To test how this would alter the strain and structure of the tumor microenvironment, we simulated progressive strain-dependent fiber deposition. A circular region of 5,432 alveolar wall elements surrounding the tumor were included in the computation for possible thickening (S4A Fig). Within this region, we increased the width of shell elements from 10 μm to 30 μm in alveolar walls that reached a threshold strain value. We used a strain threshold of 28% Lagrange strain at 20% applied stretch, which is just above the maximum strain experienced in the normal lung model without a tumor (Fig 3D). 99.9% of normal lung alveolar wall elements in the normal lung model exhibit a strain ratio < 28% (S3A Fig). The overall strain throughout the system did not change with each iteration (S4B Fig). However, strain amplification occurred in tumor-adjacent alveolar segments and was pushed further out into the surrounding lung tissue with each iteration (S4C Fig). One round of strain amplification and fiber thickening led to incomplete thickening of the alveolar walls adjacent

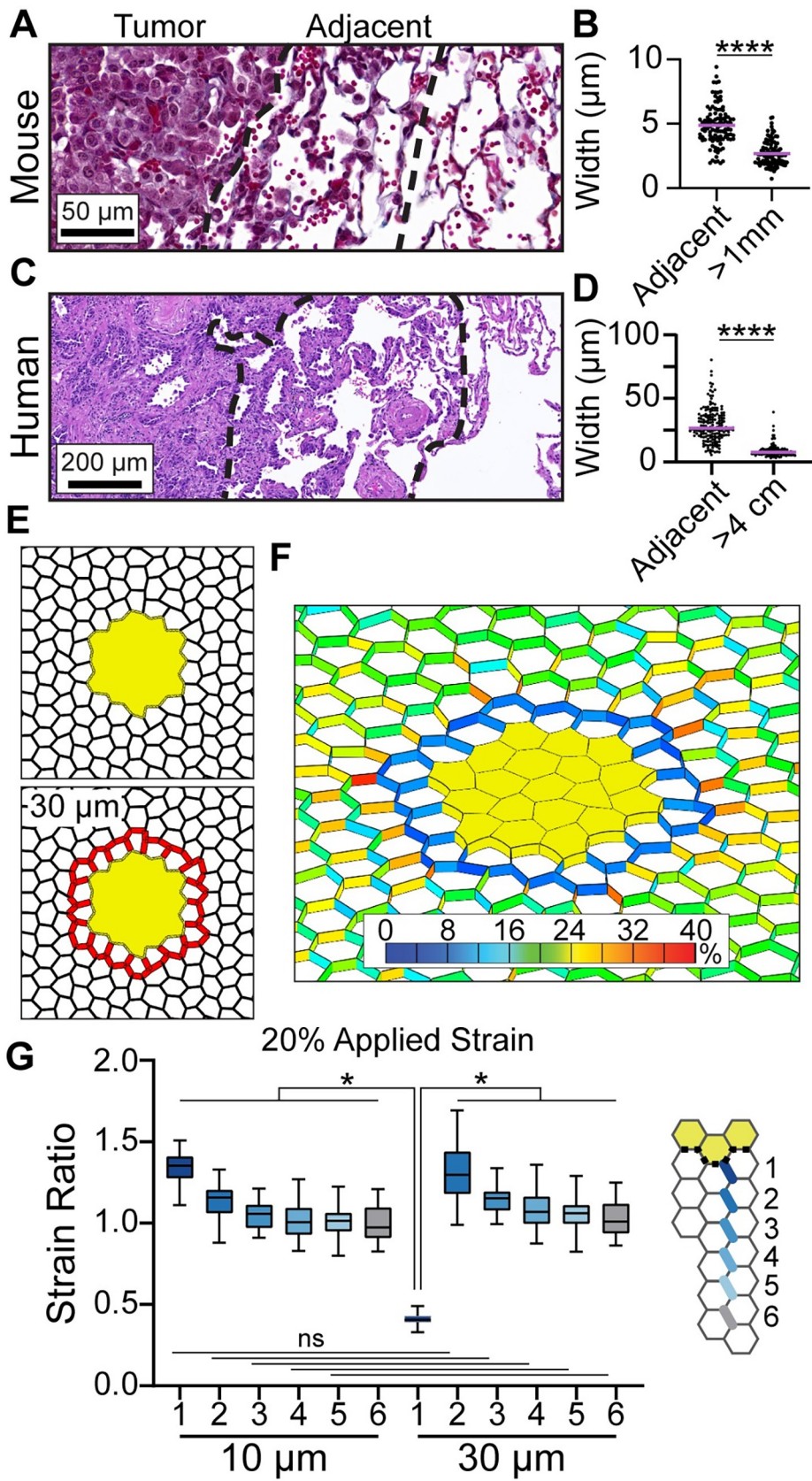

**Fig 5. Thick alveolar walls around the tumor displaces strain amplification further into the lung. A.** Early tumor in the mouse LUAD model (Masson Trichrome, original magnification 40x). **B.** Quantification of the alveolar wall width around mouse tumors within 3 alveoli from the tumor edge (adjacent) and > 1 mm away from the tumor. Measurements are pooled from 3 tumors in 2 mice. **C.** Human T1 LUAD (H&E, magnification 10x) **D.** Quantification of human alveolar wall width from 5 cases of T1 LUAD with a distinct tumor boundary and 2 samples of normal tissue >4 cm away from any tumor. **E.** Diagrams of tumor models in which adjacent alveoli have walls with original thickness of 10 μm or thickened walls of 30 μm. Thickened walls are indicated in red. **F.** Strain colormap around a 2 kPa tumor with thickened alveolar walls, 20% stretch. **G.** Strain ratio at 20% applied stretch between the baseline (10 μm) and thickened (30 μm) condition. At each alveolar distance from the tumor, $p > 0.999$ for 10 μm vs 30 μm comparisons.

to the tumor (Fig 6A). With each iteration, some of these walls were extended further out into the tumor microenvironment, creating radial tracks of thickened alveolar walls (Fig 6A). This pattern of alveolar wall thickening is reminiscent of tracks of bundled ECM fibers that are associated with tumor cell dissemination and therapy resistance in animal models of breast cancer and melanoma [55–57].

We looked for evidence of ECM tracks in clinical samples of early LUAD by staining for Tenascin-C. Tenascin-C is expressed in response to strain in fibroblast cell lines [58], organizes and bundles other ECM proteins in the tumor microenvironment [9, 59, 60], and portends poor LUAD prognosis [9]. At the invasive edge of the LUAD, we observed tracks of Tenascin-C extending into the normal tissue (Fig 6B and 6C). Tumor cells populated the tracks, indicative of dissemination into surrounding tissue (arrowheads, Fig 6D). Together, these observations suggest that strain amplification in lung cancers could contribute to structural modification of the tumor microenvironment that promotes tumor invasion.

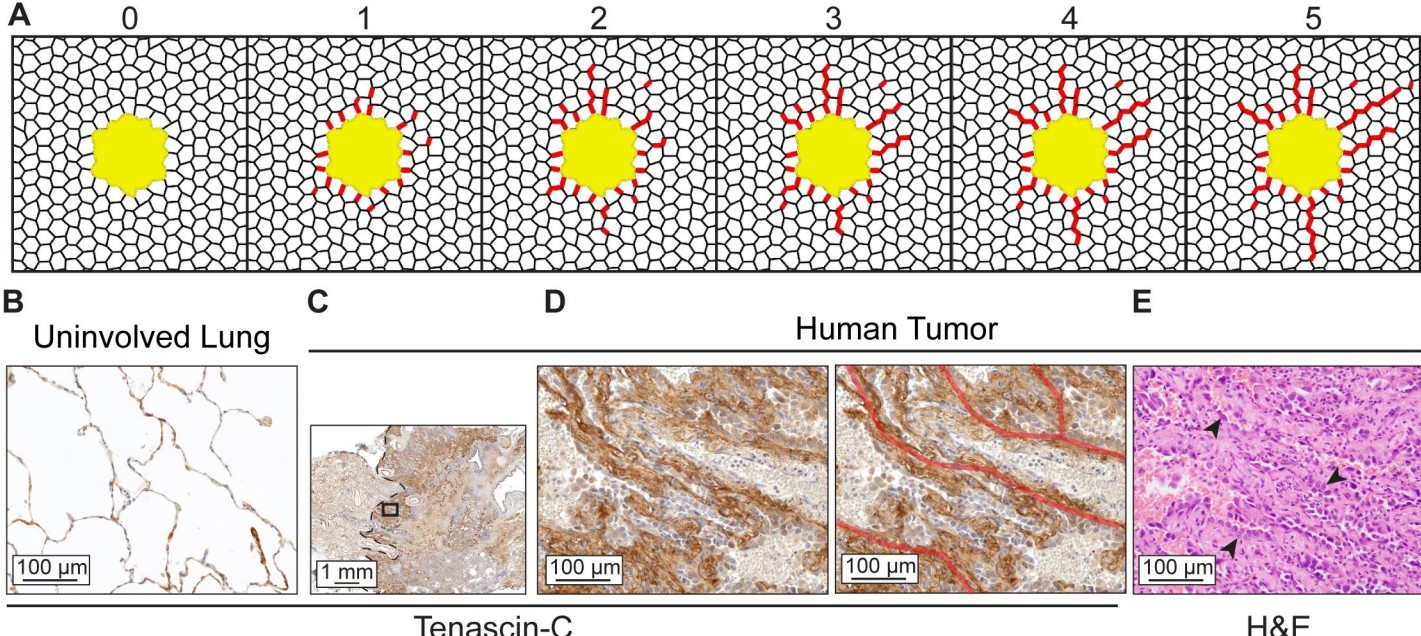

**Fig 6. Strain-mediated alveolar wall thickening produces tracks that emulate ECM remodeling in human LUAD. A.** Simulated thickening in alveolar wall elements that experience 28% Lagrange strain at 20% applied stretch for 5 iterations. Alveolar walls thickened to 30 μm are represented in red. **B.** Tenascin-C staining of a normal human lung, representative of $n = 2$. **C.** Tenascin-C staining at the edge of a T2 human tumor, representative of $n = 6$ T1 and T2 tumors. Dotted line shows the tumor edge (DAB with hematoxylin, 10x magnification). **D.** Inset from C. shows tracks of thick, Tenascin-C-stained alveolar walls that extend out from the tumor (40x magnification). Red lines mark the tracks of Tenascin-C. **E.** Serial section from C. (H&E, 40x magnification). Arrowheads indicate tumor cells with enlarged nuclei residing on thickened tracks.

## Discussion

This research interrogated how a tumor in an expanding lung could influence cellular-level strain. We incorporated a tumor into a geometric model of the lung to predict and map strains in the tumor microenvironment. Our results demonstrated that the strain experienced by alveolar walls near the tumor and at the edge of early tumors is larger than that experienced in healthy lung tissue. The model also predicted that tumor stiffening associated with tumor progression would lead to additional strain amplification that could exacerbate the altered mechanical environment. Alveolar wall thickening observed around a tumor in the lung would displace the strain amplification to the next adjacent alveolus further into the tumor microenvironment and could progressively lead to the formation of thick fiber tracks associated with tumor invasion.

Our finding of strain amplification at the earliest stage of lung cancer, before the onset of fibrotic remodeling, suggests that mechanical signaling at the tumor edge could promote early lung tumor progression. The strain amplification identified in our lung tumor model would be sensed by tumor cells at the invasive front and fibroblasts in the immediate neighborhood. We expect that some strain amplification would also occur with benign lung tumors, although to a lesser extent since benign tumors exhibit abnormal growth and thickening but retain an alveolar structure. Stretch activates cell signaling through voltage-gated ion channels and focal adhesions [61]. Epithelial cells sense and respond to the magnitude of strain energy with transcriptional induction of oncogenic behavior: cell proliferation, survival, stemness, and migration [32,62–64]. The *in vivo* consequence of strain to the disease progression of benign and cancerous lesions remains to be determined.

Fibroblasts respond to strain with increased matrix production and can remodel ECM to construct matrix tracks [54,65,66]. *In vitro*, low strains of 5–10% were found to reduce proliferation, matrix production, and myofibroblast differentiation [67], while strains at 20–30% increased fibroblast proliferation and matrix production [24,68]. This suggests that strain amplification in the lung could push the strain to a magnitude that causes fibroblasts to deposit matrix. We found that modeling matrix deposition as alveolar wall thickening pushes the strain amplification further out into the tumor microenvironment, which suggests that altered mechanical tension may create a feedforward loop of strain and matrix deposition. Such feedforward loops have been described in progressive pulmonary fibrosis, where fibrotic remodeling by activated fibroblasts further promotes fibrosis [32,69] and in glioblastoma, in which stiffness-sensing tumor cells deposit and remodel the ECM [16]. We also found that progressive strain-dependent thickening of alveolar walls can generate tracks of thick walls emanating in a radial pattern out from the tumor. Tracks of aligned ECM, including those composed of collagen, fibronectin, and Tenascin-C are a signature of invasive cancer and can serve as a conduit for tumor cell dissemination [55,57,59,70–72]. Tumor cells preferentially invade along aligned fibers and we observed tumor cells populated along Tenascin-C tracks in early human LUAD. Together, these results suggest that strain amplification in early lung cancer may initiate Tenascin-C-containing matrix deposition, which creates tracks of matrix that promote tumor cell invasion.

Our modeling approach allowed us to test how the presence of a tumor alters the magnitude and spatial distribution of strain in lung tissue. Tissue scale lung mechanics have previously been modeled using either network or constitutive models. Network lung models have employed 2- or 3-dimensional lattices of spring elements to assess strain in individual alveolar walls. The models revealed the relationships between lung pressure, volume, and alveolar stress [39] and between lung elasticity and alveolar wall stiffness and distortion [40,41,48,73]. Constitutive lung models have applied strain energy functions based on the behavior of bulk lung

tissue in uni-axial tensile tests [36–38] and the pressure-volume changes that occur at physiological stretch [43]. Our geometric model reflects the normal and tumor geometries observed in mouse models and human histopathology (Fig 1, [21]) as well as realistic material properties established in a constitutive model normal lung tissue [43].

Lung parenchyma is homogenous and isotropic [74], therefore the strain energy function derived from bulk material characterization is applicable to our lung lattice model without consideration of tissue orientation. Parameter optimization of our lung model for the range strain matching physiological respiration yielded a lung tissue modulus of 35 kPa and a linear stress-strain response. The modulus was consistent with the reported 30–50 kPa measured by micro-indentation [42]. The microindentation measurement was of decellularized lung tissue, but applied physiological stretch (~20%, [42]) to better represent native tissue. A linear stress-strain response has been demonstrated in bulk lung under physiological stretch [48–50]. However, the microindentation of decellularized lung tissue resulted in a non-linear force-displacement relationship [42]. The non-linear stress-strain response in decellularized tissue is expected, given that a stress-strain relationship solely dependent on ECM is nonlinear [41,75,76].

We assumed that the tumor was space-filling within the designated alveoli and that the alveolar walls could not be broken, did not have natural variations in width, and were constructed of homogenous material. In our model, all the alveolar walls are vertical. In the native lung, the angle of these walls vary. This geometric limitation of wall angles would influence the angles of the junctions between alveoli and thus alter the patterns of strain amplification.

While these simplifications enabled computation and the initial discoveries of strain amplification, the assumptions could be relaxed in future models to understand strain during tumor progression. A limitation of the 2.5D model, compared to the 3D lung, is that the junctions between the alveoli and the top and bottom surface of the alveolar sheet are modeled as uniformly perpendicular to the plane of the sheet. This restricts the angle of strain with respect to the tumor, and may predict more uniform tracks than might be observed with a randomized 3-dimensional geometry. In invasive LUAD, the tumor grows to occupy new alveolar spaces and invades through breaks in the basement membrane layer around the tumor [77,78]. Our assumption that the integrity of the alveolar walls is maintained around the tumor constrains the tumor size and distributes the tension throughout the edge of the tumor. In normal lung, alveolar walls exhibit variable widths in which thin walls exhibit more strain than thicker walls [45]. Such heterogeneity could lead to even greater strain amplification in alveoli adjacent to the tumor when the alveoli are comprised of thinner walls. Highly elevated strain could lead to alveolar wall rupture, observed in histological sections (Fig 1, [79]), and would change the local strain distribution. In our model, we assumed the thickened alveolar walls adjacent to a tumor had the same material properties as the regular width alveolar wall. However, oncogenic remodeling, including the deposition of new ECM proteins, ECM alignment, and crosslinking, likely stiffens the alveolar walls [33]. Thus, our model likely underestimated the impact of alveolar wall remodeling around the tumor.

The identification of tumor-induced strain amplification in the tumor microenvironment underscores the importance of future research into strain and mechanotransduction in lung cancer. If strain amplification promotes ECM deposition and tumor invasion, signatures of strain-sensitivity could serve as a new marker to distinguish indolent from aggressive early lesions and strain-induced ECM remodeling may be a new therapeutic target for blocking progression.

## Supporting information

**S1 Fig. Parameter optimization for lung lattice.** Force-Displacement relationship of the Birzle, et al. constitutive model [43] and the neo-Hookian approximation (Modulus = 35 kPa,

Poisson Ratio = 0.25). The assessed range of displacement values reflected physiological strains. Displacement of 15,000 μm = 50% applied stretch. Displacement of 15,000 μm = 50% applied stretch.
(TIF)

**S2 Fig. Calculated lung modulus by indentation agrees with experimental AFM results. A.** Schematic of simulated indentation. A spherical rigid body represents the AFM tip and a solid volume matches the Birzle, et al. constitutive model [43]. The rigid body is displaced vertically at an indentation depth (d) equivalent to the sphere radius (r), 2.5 μm. **B.** Visualization of the rigid sphere and associated reaction force in the initial position above the material and **C.** after displacement into the material.
(TIF)

**S3 Fig. Strain ratio increases with tumor modulus. A.** Distribution of alveolar wall strain in the normal lung model (no tumor) under 20% applied stretch. The 8 layers of alveoli around the edge of the modeling domain were excluded from the calculations to avoid edge effects. Median Lagrange shell strain is 20.51%, which corresponds to a strain ratio of 0.93. **B.** Strain ratio in the tumor-adjacent alveolar walls (Fig 3D–3F wall 1). Boxes are 25[th] percentile to 75[th] percentile with the median marked by the central line. Red lines show the median Lagrange shell strain for the 2 kPa and 20 kPa tumors at 20% stretch (29.8% and 32.5%), which correspond to strain ratio 1.4 and 1.5. **C.** Relationship between tumor modulus and strain ratio for applied stretch values of 5, 20, and 50%. The strain ratios for large tumor modulus values converged at 1.7 with 50% stretch, 1.5 with 20% stretch, and 1.3 with 5% stretch.
(TIF)

**S4 Fig. Overall strain in the system redistributes during strain-mediated alveolar wall thickening. A.** Region of interest including 5,432 elements selected for analysis around the tumor includes ~15 layers of alveoli. **B.** Distribution of Lagrange shell strain in all selected elements in the normal lung and after each simulation iteration of 20% applied stretch and shell-thickening in Fig 6A. **C.** Colormaps of Lagrange strain in alveolar walls around the tumor during strain-mediated thickening. Lattice is positioned to match the orientation of the tumor in Fig 6A.
(TIF)

**S1 Table. Mesh Convergence Values.**
(DOCX)

## Acknowledgments

We thank the FEBio development team for their continued efforts to develop, improve and support the FEBio software.

## Author Contributions

**Conceptualization:** Rebecca G. Zitnay, Michelle C. Mendoza.

**Formal analysis:** Rebecca G. Zitnay.

**Funding acquisition:** Jeffrey A. Weiss, Michelle C. Mendoza.

**Investigation:** Rebecca G. Zitnay, Michelle C. Mendoza.

**Methodology:** Rebecca G. Zitnay, Michael R. Herron, Jeffrey A. Weiss, Michelle C. Mendoza.

**Project administration:** Michelle C. Mendoza.

**Resources:** Scott Potter, Lyska L. Emerson.

**Software:** Michael R. Herron, Keith R. Carney, Jeffrey A. Weiss.

**Validation:** Scott Potter, Lyska L. Emerson.

**Visualization:** Lyska L. Emerson.

**Writing – original draft:** Rebecca G. Zitnay.

**Writing – review & editing:** Lyska L. Emerson, Jeffrey A. Weiss, Michelle C. Mendoza.

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
