## [Decision Letter · Decision Letter 0]

18 Jul 2022

Dear Dr. Mendoza,

Thank you very much for submitting your manuscript "Mechanics of lung cancer: A finite element model shows strain amplification during early tumorigenesis" for consideration at PLOS Computational Biology. As with all papers reviewed by the journal, your manuscript was reviewed by members of the editorial board and by several independent reviewers. The reviewers appreciated the attention to an important topic. Based on the reviews, we are likely to accept this manuscript for publication, providing that you modify the manuscript according to the review recommendations.

Sincerely,

Philip K Maini

Associate Editor

PLOS Computational Biology

Feilim Mac Gabhann

Editor-in-Chief

PLOS Computational Biology

[LINK]

Reviewer's Responses to Questions

**Comments to the Authors:**

Reviewer #1: The problem under investigation is how the increased strain due to respiration in a lung tissue with a cancer tumor could promote tumor progression. A finite element analysis is performed and both, the computer simulations and the results, use experimental and/or clinical data that are either available in the literature or presented in this manuscript. The experiments and simulations appear to be well done and the manuscript is generally well-written. The performed study and results are very interesting and thus I support the publication of the manuscript after the following minor issues are addressed.

1. A brief review of the literature on computational models of lung mechanics and lung tumors should be added to the Introduction. The novel contribution of this work to the field should be better highlighted by clearly differentiating it from previous publications, especially reference [37].

2. Why is FEBio software chosen for this study? Please provide the most relevant feature(s) of this software that contributed in deciding to use it here.

3. In the subsection titled 'Description of the finite element models of lung tissue', the following statement is made: 'To account for variability in lung architecture, a randomization algorithm was applied to the initial geometry, followed by simulated annealing through energy minimization [38]'. How does this work exactly? A very brief description of this process (either in the text or as a footnote) would help.

4. From the text, it is not clear which of the two strain energies (1) or (3) are used to model the healthy lung tissue and the tumor. Is the so-called Birzle hyperelastic constitutive model (1) used for the lung tissue without tumor and for the lung tissue surrounding the tumor, while the compressible neo-Hookean hyperelastic constitutive model (3) is used only for the tumor? Or model (3) is used to model the mechanical behavior of both lung tissue and tumor in the case that the tumor is present? Please clarify and make necessary edits to the test accordingly.

5. What boundary conditions were used on the lung tissue without a tumor? What extra boundary conditions were imposed in the case of the lung tissue with a tumor at the interface between the tumor and the surrounding tissue?

6. The statistics was calculated with the Andersen-Darling, nonparametric Kruskal-Wallis, and post-hoc Dunn’s multiple comparisons tests. What are the most important features of these tests that contributing in deciding to use them here? (This could be a footnote.)

7. How do Figs. 6 D-E compare to Fig. 6 A? Highlighting the tracks mentioned in the caption of Fig.6 on the images in Fig. 6 D-E with a marker (instead of just showing stars and arrows) might help the comparison with the finite element results shown in Fig. 6 A.

8. Are there any special characteristics that a cancerous lung tumor has that makes this study work only for these tumors and not for benign ones? For instance, how do the cells geometry look in a cancerous lung tumor and its surrounding tissue? Are there any differences seen in the cells geometries in a healthy lung tissue versus a lung tissue with a cancer tumor present? How do these shapes compare to those for a benign tumor? Given the diffusive character of cancer cells that makes them so invasive, could a connection between the tracks shown in Fig. 6A and the possible diffusion paths taken by the cancer cells be established? Could this be expected to happen in the case of a benign tumor? Lastly, could it be possible that at a certain strain threshold, a certain mechanotransduction process be activated that will transform a benign tumor into a cancer one? Some of these questions go beyond the work presented here so they do not need to be addressed.

Reviewer #2: This paper presents a computational analysis of strain-related alterations to lung tissue due to tumour growth. The authors use an established constitutive model for lung tissue and implement a FE computational model to simulate tissue stiffening due to increased strain around stiff tumours. The simulations demonstrate how strain-related tissue stiffening and wall thickening can lead to stiff tracks radiating from the tumour, providing "highways" for tumour cell migration. The work is thus relevant as a demonstration of this process, and their results are related to imaging observations from mouse and human lung tumours. Overall I have no major concerns for this work and recommend the paper for publication with minor revisions

The comments I have for the authors are below:

- how do they implement the stretching of the tissue, i.e. did all boundaries move or just top/bottom?

- would the choice of geometry affect the results if a radial geometry was used instead of a cartesian setup?

- does the tissue experience any shear around the tumour?

- is lung tissue generally isotropic or anisotropic? Are there implications of this assumption on their results?

- what are the limitations of the 2D model compared to the 3D lung?

- should equation 5 be solved for E if that is the formula used to compute E from F?

**Have the authors made all data and (if applicable) computational code underlying the findings in their manuscript fully available?**

Reviewer #1: Yes

Reviewer #2: None

PLOS authors have the option to publish the peer review history of their article (what does this mean?). If published, this will include your full peer review and any attached files.

Reviewer #1: No

Reviewer #2: No

Figure Files:

Data Requirements:

Reproducibility:

References:

---

## [Decision Letter · Decision Letter 1]

6 Oct 2022

Dear Mendoza,

We are pleased to inform you that your manuscript 'Mechanics of lung cancer: A finite element model shows strain amplification during early tumorigenesis' has been provisionally accepted for publication in PLOS Computational Biology.

Best regards,

Philip K Maini

Academic Editor

PLOS Computational Biology

Feilim Mac Gabhann

Editor-in-Chief

PLOS Computational Biology

Reviewer's Responses to Questions

**Comments to the Authors:**

Reviewer #1: I recommend the revised manuscript for publication.

Reviewer #2: The authors have satisfactorily addressed my concerns and I recommend the article for acceptance and publication.

**Have the authors made all data and (if applicable) computational code underlying the findings in their manuscript fully available?**

Reviewer #1: Yes

Reviewer #2: Yes

PLOS authors have the option to publish the peer review history of their article (what does this mean?). If published, this will include your full peer review and any attached files.

Reviewer #1: No

Reviewer #2: No

---

## [Editor Report · Acceptance letter]

18 Oct 2022

PCOMPBIOL-D-22-00665R1 

Mechanics of lung cancer: A finite element model shows strain amplification during early tumorigenesis

Dear Dr Mendoza,

I am pleased to inform you that your manuscript has been formally accepted for publication in PLOS Computational Biology. Your manuscript is now with our production department and you will be notified of the publication date in due course.

With kind regards,

Anita Estes
